# Actin and Myosin in Non-Neuronal Exocytosis

**DOI:** 10.3390/cells9061455

**Published:** 2020-06-11

**Authors:** Pika Miklavc, Manfred Frick

**Affiliations:** 1School of Science, Engineering & Environment, University of Salford, Manchester M5 4WT, UK; 2Institute of General Physiology, Ulm University, Albert-Einstein-Allee 11, 89081 Ulm, Germany

**Keywords:** secretion, vesicle trafficking, cell cortex, actin coat

## Abstract

Cellular secretion depends on exocytosis of secretory vesicles and discharge of vesicle contents. Actin and myosin are essential for pre-fusion and post-fusion stages of exocytosis. Secretory vesicles depend on actin for transport to and attachment at the cell cortex during the pre-fusion phase. Actin coats on fused vesicles contribute to stabilization of large vesicles, active vesicle contraction and/or retrieval of excess membrane during the post-fusion phase. Myosin molecular motors complement the role of actin. Myosin V is required for vesicle trafficking and attachment to cortical actin. Myosin I and II members engage in local remodeling of cortical actin to allow vesicles to get access to the plasma membrane for membrane fusion. Myosins stabilize open fusion pores and contribute to anchoring and contraction of actin coats to facilitate vesicle content release. Actin and myosin function in secretion is regulated by a plethora of interacting regulatory lipids and proteins. Some of these processes have been first described in non-neuronal cells and reflect adaptations to exocytosis of large secretory vesicles and/or secretion of bulky vesicle cargoes. Here we collate the current knowledge and highlight the role of actomyosin during distinct phases of exocytosis in an attempt to identify unifying molecular mechanisms in non-neuronal secretory cells.

## 1. Introduction

Regulated secretion is a fundamental process in many different types of eukaryotic cells. In general, vesicle contents are released by exocytosis of secretory vesicles in response to an extracellular stimulus. This involves a sequence of highly regulated steps. Secretory vesicles are transported to the plasma membrane (PM), fuse with the PM and release their secretory cargo into the extracellular space. The basic principles are conserved amongst all secretory cells, mainly neurons, hematopoietic, endocrine and exocrine cells [1]. However, differences in vesicle size and physical and functional properties of secreted molecules require specific adaptations to facilitate efficient secretion. 

In particular, cytoskeletal elements such as actin and myosin have been found to play distinct roles within different types of secretory cells. Historically, it was believed that the cortical actomyosin complex beneath the PM mainly acts as a mechanical barrier to prevent “premature” fusion of secretory vesicles with the PM. This hypothesis mainly originated from early observations in neuronal and neuro-endocrine cells. The recent development in imaging technologies and the wider interest in secretion from non-neuronal cell types, however, have revealed a much wider impact of the actomyosin system on secretion. Novel molecular pathways and regulatory mechanisms of the actomyosin system have been described. It is well established now that the actin cytoskeleton provides tracks for trafficking of secretory granules towards the fusion sites [2,3] and the cortical actomyosin network is involved in vesicle anchoring, docking and fusion with the PM [4]. Following fusion, during the post-fusion stage, the actomyosin network facilitates secretion and/or retrieval of vesicle membrane from the PM. Some of these functions have been first described in non-neuronal cells and often reflect adaptations to the exocytosis of large secretory vesicles and the release of bulky and/or hydrophobic cargoes. Within this review, we aim to summarize recent findings and current knowledge on the role of the actomyosin system for exocytosis and secretion in non-neuronal exocytosis. We collate the current knowledge obtained from cells with large secretory vesicles and/or bulky vesicle contents. We specifically highlight the role of cytoskeletal elements during distinct phases of vesicle exocytosis in an attempt to identify unifying molecular mechanisms across these secretory cells (Figure 1) and provide a list of central cytoskeletal elements and regulatory molecules associated with actin that have been identified so far (Table 1). For the role of the actomyosin complex in neuronal and electrically excitable cells (e.g., neuro-endocrine cells), the reader is referred to excellent recent reviews [4,5,6,7,8,9].

## 2. Vesicle Transport

Biogenesis of secretory vesicles typically starts in the center of the cell, within the Golgi network. Lipids and proteins destined for secretion or insertion into the PM are packed in vesicles that bud off the trans-Golgi network. The vesicles are then transported (trafficked) to the cell periphery to finally fuse with the PM. The cytoskeleton provides intracellular tracks to facilitate directional movement of secretory vesicles towards PM. In mammalian cells, microtubules serve as major tracks for long distance vesicle transport by the motor proteins kinesin and dynein. Thus microtubule organization is important for targeted delivery of secretory carriers to the cell periphery [10,11]. Binding to actin helps to move vesicles that are close to the cell cortex [4,12,13].

In mammalian pancreatic acinar cells [14] and *Drosophila* larval epithelial cells [15], organized bundles of actin filaments have been observed in addition to the cortical actin network. These actin bundles protract towards the cell interior and are attached to the PM via Dia, a member of the formin nucleation factor family that produces linear actin filaments. Secretory vesicles are moved along these bundles to the apical membrane. This is accomplished by myosin V (myoV), which regulates short-range transport close to cell apical membrane. When these tracks were disrupted, apical transport of vesicles was hampered [14,15,16].

MyoV is the best-known actin-associated molecular motor involved in transport of secretory vesicles towards the PM [17,18]. MyoV is a processive molecular motor with two motor heads and a tail domain. One of the two motor heads is associated with actin during the myosin motor cycle and alternate binding of both heads to actin results in myosin “walking” along actin tracks. Actin filaments are polarized and myoV is directed towards the barbed end of the actin filament. There are three different class V myosins in vertebrates (myoVa, b, c) and myoVa is most often the one implicated in secretory vesicle trafficking, although a role for myoVb and c are also emerging in recent literature [19,20]. MyoVa can transport vesicles because its tail domain can be associated with a vesicle via diverse molecular linkers on secretory vesicle membranes [3,21]. 

Vesicle membrane proteins that can provide the connection to myosin motors are Rab GTPases. Rab proteins are a large family of small GTPases that selectively bind to different intracellular compartments involved in trafficking [22,23]. Traditionally, Rab3 and Rab11 GTPase isoforms were associated with the secretory pathways; however, recently it is becoming clearer that other Rab GTPases such as Rab27 also play an important role [24]. In transgenic mice expressing EGFP-Rab27a, it was shown that Rab27a was expressed in a wide variety of secretory cells, including exocrine, endocrine and hematopoietic cells [25]. The functional significance of Rab27a localization to secretory vesicles in non-neuronal cells was described in endothelial cells [26], oocytes [27], sperm [28], melanocytes [29] and cytotoxic T (CT) lymphocytes [30].

MyoV binds to secretory vesicle associated Rab indirectly via linker molecules. In endothelial cells, which secrete high molecular weight protein von Willebrand factor (vWF) from cigar-shaped secretory vesicles termed Weibel–Palade bodies (WPBs), the linker molecule between myoVa and Rab27a is the Rab effector MyRIP (myosin and Rab27a-interacting protein, also named exophilin 8) [13,26]. MyRIP can also directly bind to actin and this interaction is important for regulation of WPB exocytosis [12]. In melanocytes, myoVa is bound to Rab27a via another Rab effector melanophilin [31]. 

Together these findings suggest that transport mediated by actin and myoV functions via a conserved set of accessory proteins. In most cases actin provides tracks for short-range transport near the cell periphery. However, the actin cytoskeleton can also provide a means for long-range trafficking. Disruption of microtubules with nocodazole had no effect on cortical granule trafficking from the cell center towards the PM in oocytes. However, when the actin network was disrupted with cytochalasin, vesicle transport was abolished, suggesting a pivotal role for actin in transport [27,32]. 

Following transport to the PM, a myoV-mediated connection between actin and secretory granules was also suggested to be important for vesicle attachment to the cell cortex [18].

## 3. Docking

Docking and fusion of secretory vesicles with the PM require a cortical actin network and its remodeling [4]. Cortical actin is a dense network of actin filaments attached to the PM. It contains more than 150 different actin binding proteins involved in actin nucleation, crosslinking and contraction [33]. Actin fibers are attached to PM with ezrin/radixin/moesin proteins and myosin I (myoI) family molecular motors [34,35]. Cortical actin is mostly a very thin structure (100 nm), although the thickness can increase up to 4 μm in some specialized cells such as oocytes [33,36]. 

Formins and Arp2/3 represent the most important actin nucleators in the actin cortex. Their contribution to production of actin filaments in the cell cortex varies in different cell types and in different stages of cell division [33]. This affects actin network structure because formins produce linear actin filaments, whereas Arp2/3 nucleates branched filaments [37,38]. Action of different actin nucleators creates variability in cortical actin density and turnover, which are important parameters influencing secretory granule exocytosis. Cortical actin density measurement with atomic force microscopy suggested a mesh size of 100 nm or less beneath the PM [39]; however, actin density may differ through cortex cross-section [40]. Cortical actin turnover kinetics is not well understood but it is likely on the order of seconds to minutes [33,39].

Due to its density, cortical actin was long regarded as a diffusion barrier that prevents access of granules to secretory sites and traps secretory vesicles in the cortical network [41,42]. In non-neuronal secretory cells, the role of cortical actin for exocytosis was mostly investigated by chemical disruption of actin cytoskeleton. Actin depolymerization with cytochalasin E resulted in more fusion events in stimulated endothelial cells, which suggested an inhibitory role of cortical actin [43]. Similarly, actin depolymerization with latrunculin B [44] or by knock-down of formin mDia1 [45] resulted in increased secretion in mast cells. On the other hand, stabilization of actin filaments with jasplakinolide inhibited exocytosis in natural killer (NK) cells [46]. FRAP experiments showed increased apical cortical actin turnover in lacrimal acinar cells stimulated for secretion compared to control [47]; however, in other cell types general changes in cortical actin after secretion stimulation were not observed [48]. 

Actin cortex remodeling is necessary to enable access of large secretory vesicles to the PM for fusion of vesicles with the PM. Myosin II (myoII), which is ubiquitously present in the cortical network, was suggested to play a pivotal role in this remodeling step. In NKcells, lytic granules, which contain pore forming proteins and proteases such as perforin and granzyme, are secreted at the site of immunological synapse. A recent super-resolution microscopy study showed that displacement of actin filaments created clearances in cortical actin that were sufficiently large to allow passage of lytic granules [46]. An Arp2/3 inhibitor inhibited exocytosis, suggesting that nucleation of branched filaments is necessary for cortical actin remodeling. Treatment with the myoII inhibitor blebbistatin increased the density of cortical fibers and resulted in a lower number of clearances and diminished exocytosis, which together suggests a dual mechanism for cortical actin dynamics [46]. Disassembly of the actin cortex at the site of granule secretion and involvement of Arp2/3 and myoII in cortical actin dynamics was also observed in mast cells using total internal reflection fluorescence structured illumination microscopy (TIRF-SIM) [49]. MyoII function can be promoted by molecular chaperons such as UNC-45A, which was shown to increase myoIIA binding to actin and exocytosis in NK cells [50]. Secretion of renin in the kidneys could be negatively regulated by myosin phosphorylation, because inhibition of myosin light chain kinase (MLCK) increased renin secretion [51]. MLCK phosphorylates light chains of myoII and thereby promotes actin-myosin contractility. In addition to myoII, the unconventional myoI family was also shown to play a role in exocytosis. Myo1c acts in membrane remodeling, which enhanced exocytosis of insulin-dependent GLUT4 vesicles [52]. Likewise, myo1c inhibition reduced the rate of lamellar body exocytosis in surfactant-secreting alveolar type II (ATII) cells from the lungs [53]. In myo1g deficient B lymphocytes, secretion of TNF-α and prolactin was inhibited, suggesting involvement of myo1g in membrane tension and consequently exocytosis [54].

Regulation of cortical actin reorganization is often connected to changes in PM phosphatidylinositol 4,5-bisphosphate (PIP_2_) levels and to activity of small GTPases of the Rho family. PIP_2_ regulates adhesion between the PM and cytoskeleton by affecting numerous actin-associated proteins necessary for actin crosslinking and turnover [55,56]. The role of Rho and other small GTPases for cortical actin re-organization during exocytosis has been extensively studied in neuroendocrine cells and neurons [57,58,59] but is less well researched in non-neuronal cells. It is likely that many additional factors are also involved in cortical actin remodeling. In airway mucus-secreting cells it was shown that the N-terminal domain of MARCKS (myristoylated alanine-rich C-kinase substrate) inhibits 90% of mucus secretion [60,61]. MARCKS is a PIP_2_ and actin-binding protein phosphorylated by protein kinase C (PKC). Increased intracellular Ca^2+^ concentration activates PKC, which phosphorylates MARCKS and thereby induces its translocation from the PM to the cytosol. MARCKS translocation from the PM can decrease actin filament crosslinking in the cell cortex and increase PIP_2_ availability, which facilitates cortical actin dynamics during exocytosis [62].

Lately, it has also been appreciated, that the cortical actin network is essential for directing exocytic vesicles to the fusion site and for stable attachment of granules close to the PM [41,63]. It has been suggested that the actin cortex physically facilitates exocytosis by caging vesicles at the fusion site or crosslinking vesicles to the PM and provides a scaffold on which various regulators of exocytosis are recruited, thereby orchestrating the final stage of exocytosis [4,64].

Vesicle docking was originally described as proximity of secretory vesicles to the PM on electron microscopy images or as restriction of vesicle movement close to the PM measured by TIRF microscopy. However, different methods and interpretations resulted in some discrepancy in use of this term [65]. Among numerous proteins implicated in docking, some are also involved in regulation of the actin cytoskeleton, most notably Rab proteins (Rab3/Rab27), Rab effectors (such as granuphilin and MyRIP) and myoV. The Rab27a and Slp4a (Rab27a/Rab3 effector)-mediated docking step was necessary for secretion in endothelial cells [13,26]. In addition, myoVa inhibition resulted in accumulation of WPBs in the perinuclear area and secretion of immature vWF, suggesting that secretory granule attachment to peripheral actin and regulating short term trafficking prevents premature exocytosis [3]. MyoVa was also necessary for docking of secretory granules in the MIN6 beta cell line [66]. In pancreatic islet cells from MyRIP deficient mice, there were less granules at the cell periphery and insulin secretion was decreased. Silencing experiments suggested that MyRIP anchors granules to cortical actin by binding to another myosin family member myosin VIIa (myoVIIa) and to RIM-BP protein, which is otherwise associated with presynaptic active zone structure [67]. However, in contrast to endothelial cells, direct interaction between MyRIP and myoVa was not confirmed [68]. Distinct roles for granule docking have been proposed for Rab27a and Rab27b in mast cells. Rab27b was shown to mediate vesicle docking via a Munc13-4 interaction, whereas the Rab27a/melanocortin/myoVa complex regulated granule binding to actin cytoskeleton and cortical actin rearrangements [44]. In lysosome exocytosis, which mediates PM repair after injury, Rab3a and myoII were identified as proteins involved in lysosome positioning in the actin cortex, whereas silencing of Slp4a resulted in lysosome clustering in the peri-nuclear region [69]. Lysosomal associated membrane protein 1 (Lamp1) was also suggested to mediate lysosome docking and movement towards the PM. Lamp1 was associated with myosin XI and silencing of either protein decreased lysosomal exocytosis [70].

Additional proteins associated with secretory granules were also found to be important for exocytosis in different cellular systems, although the mechanism of their function is not always clear. Annexin A7, a protein that binds to Ca^2+^ and associates with phospholipids and actin, was suggested to facilitate secretory vesicle exocytosis in ATII cells. Annexin A7 increased its association with secretory vesicles after the cells were stimulated for secretion and this depended on protein phosphorylation [71]. Similarly, annexin A2 was described to play a role in secretion in endothelial cells [72,73]. In endothelial cells, super-resolution microscopy also showed co-localization of zyxin, which is otherwise best known for its role in regulating actin polymerization at cell–matrix adhesions [74], with about 50% of vesicles before secretion and the percentage increased after vesicle fusion (75). Zyxin silencing in endothelial cells suppressed cAMP-mediated vWF secretion, but not Ca^2+^ mediated secretion [75]. Zyxin function was regulated by phosphorylation and only phosphorylated zyxin mediated secretion [75]. The importance of protein phosphorylation for secretion was also demonstrated in neutrophil granule secretion where TNF-α stimulation resulted in phosphorylation of a large number of proteins, many of which were associated with the actin cytoskeleton [76]. In mast cells, inhibition of Rab11, which was localized to exocytic vesicles, decreased exocytosis, whereas subsequent inhibition of actin polymerization restored exocytosis. It was concluded that cortical actin depolymerization necessary for exocytosis was regulated by Rab11 [77].

Finally, fusion of vesicles with the PM and opening of the fusion pore is achieved by SNARE proteins that supply the energy to overcome the high kinetic barrier to mediate lipid bilayer fusion [78]. The actin cortex is involved in regulating the fusion pore and providing the force to complete fusion [4]. 

## 4. Fusion Pore

Regulation of fusion pore dilation/closure has been identified as a key regulator for fine-tuning vesicle content secretion [79]. Specific techniques, such as content photobleaching [80,81,82], can be used to assay whether a pore is open or closed, and, consistent with earlier electrophysiological data [83], revealed that fusion pores dynamically open and close [84].

It is well established that fusion pore expansion is regulated by Ca^2+^ [85]. F-actin [82] and myoII [86,87,88] have been suggested as molecular mediators for Ca^2+^-dependent fusion pore transitions in exocytosis of large vesicles in non-neuronal cells [89,90]. Whether actomyosin provides force around the fusion pore to stabilize it or actively modulates fusion pore diameters is yet to be determined. One of the first reports came from pancreatic acinar cells that showed that F-actin keeps the fusion pore open [91]. It was later proposed that F-actin stabilizes an open fusion pore and provides a possible site of regulatory control of pore dynamics through a motor protein [82]. Subsequent reports indeed found that myoII phosphorylation directly regulates fusion pore opening in chromaffin cells [86,87] and in epithelial cells [88], leading to the conclusion that myoII acts with F-actin, to stabilize the open fusion pore [92]. In line with this, it was suggested that loss of subcortical, formin-nucleated, linear F-actin affects the stability of the fusion pore during insulin granule exocytosis in pancreatic β cells, leading to uncontrolled pore dilation and granule collapse [48]. The exact role of myoII and F-actin are still elusive. MyoII could directly interact with F-actin in a force generating manner to stabilize the pore. Alternatively, the F-actin ring around the neck of the fused granule could counteract myoII-induced membrane tension that pulls the pore open [93]. A recent report found that Rab3a is involved in regulation of fusion pore expansion in dense core granule exocytosis during the acrosomal reaction in sperm [94], However, whether this is linked to cortical actin needs to be determined. 

## 5. Post Fusion

In the classical model of secretion, the initial fusion pore dilates until the membrane is completely collapsed into the PM and the entire vesicle content is passively released in an all-or-none manner (‘full fusion’) [95,96]. However, this view started to change when findings from neuronal and non-neuronal cells suggested that fused secretory vesicles do not passively collapse into the PM due to an increasingly widening fusion pore but shrink while maintaining the “omega shape” [97,98,99,100]. The cell cytoskeleton seems to play a pivotal role in this process and a multitude of studies within the last decade focused on the importance of actin and myosin action during the post-fusion phase for regulating secretion and retrieval of excess PM. Although experimental studies suggest parallels between mechanisms operating in neuronal and non-neuronal cell types [101,102,103], we focus here on non-excitable cells.

### 5.1. Vesicle Content Release

In many secretory cells actin and myosin are specifically recruited to the surface of secretory granules following their fusion with the PM. Actin and myoII coating of fused granules had already been observed several decades ago [104,105]. Early studies suggested a rather passive role of the actomyosin coat, stabilizing the limiting membranes of fused secretory granules to facilitate vesicle content release [106,107]. Since, it has become clear that actomyosin-dependent compression of fused granules is essential to actively drive secretion in many secretory cells. This has been shown for secretion of tear proteins from rabbit lacrimal acinar epithelial cells [47], vWF from endothelial cells [43], proteins from acinar cells in salivary glands [108], pulmonary surfactant from alveolar epithelial cells [80], salivary cells in *Drosophila* larvae [64] and insulin secretion from pancreatic beta cells [48]. All these cells contain comparatively large secretory vesicles that are in the µm range rather than nm-sized vesicles found in neurons and neuroendocrine cells.

Most studies focused on a specific set of molecules/regulators of actin coat formation and compression in different secretory cells. However, some overlap exists between individual studies and fundamental principles seem to be widely conserved. Membrane mixing seems a prerequisite for actin coat formation. Upon fusion, key signaling molecules of the PM can diffuse into the fused secretory granule membrane and act as trigger for local actin assembly (‘kiss-and-coat’) [109]. In particular, phosphoinositide signaling has been found in several species. PIP_2_ diffuses from the PM into the membrane of fused vesicles in salivary glands of *Drosophila* [64] and in ATII epithelial cells (unpublished observation) and diacylglycerol (DAG), a product of PIP_2_ hydrolysis by phospholipase C, was identified as the main regulator of actin coat formation on cortical granules in *Xenopus* oocytes [110]. Subsequently, Rho GTPases are recruited/activated on fused secretory vesicles to initiate Rho-dependent actin polymerization. Rho GTPase recruitment/activation was either observed directly [80,106,111] or deduced from impaired actin coat formation upon inhibition of Rho GTPases [107,112]. Nucleation of actin filaments is then mediated by formins [48,80,111] or the Arp2/3 complex [41,95]. Given the observed dynamics of actin coat formation it has also been speculated that a yet unidentified rapid nucleating system may be involved [113]. 

Formins have been found to be necessary for formation of actin coats within numerous secretory cells [48,80,111]. Arp2/3, on the other hand, does not seem to be essential for initial formation of the actin coat in most cells [43,80,111] apart from a central role in insulin secreting beta cells [48]. In a *Drosophila* line expressing fluorescently labelled Arp3, Arp3 was recruited to vesicles with considerable delay (~29 s) after actin coat formation [64]. Arp2/3 is probably involved in coat contraction within some secretory cells [48,64], but not in others [43,80].

It is tempting to speculate on whether the main nucleation mechanism determines subsequent coat compression and/or stabilization. Formins mediate nucleation of unbranched, linear filaments [37] and accelerate elongation of the fast-growing (barbed) ends of actin filaments [114,115,116]. The Arp2/3 complex initiates branched actin filament formation by binding to the sides of pre-existing filaments serving as a template for elongation [38]. This might be important because different mechanisms for coat contraction have been described.

A role for myoII in actin coat contraction has been reported in most systems [43,47,80,107,108,117]. However, myoII activity is not the main driving force for actin coat contraction. In all systems investigated so far, inhibition of myoII activity decreased vesicle compression rates, but did not prevent coat contraction [43,47,80,88,107,108,117,118]. MyoII recruitment usually follows coat formation [64,117,119,120] and it was suggested that myoII recruitment also depends on actin coat formation [111]. However, recent studies show that recruitment of myoII A (NMIIA) filaments is F-actin independent [119,121] and imply the existence of an alternate NMIIA receptor on the secretory granule surface [122]. MyoII recruitment/activation is likely dependent on Rho-associated protein kinase (ROCK) and/or MLCK [111,119,120]. Some studies even suggested distinct roles for myoII isoforms [119,122]. Unconventional myosins are also recruited to actin coats. Myo1c is recruited to actin coats on secretory vesicles in ATII cells [53] and on cortical granules in *Xenopus* oocytes [123]. Inhibition of myo1c inhibited actin coat contraction. It was proposed that myo1c is necessary to attach the actin coat to the vesicle membrane [123]. Myo1b [53] and myo1e [117] also translocate to actin coats. Their inhibition affected actin coat contraction or formation, respectively.

It has also been speculated that actin polymerization alone might be sufficient to compress the exocytic vesicle [106,124]. Recently, we showed that regulated actin depolymerization and actin fragment crosslinking drive coat contraction. Actin depolymerization by cofilin-1 caused filament sliding and coat contraction in the presence of a cross-linker, α-actinin. Contraction was independent of motor activity and actin filament organization [120]. Deletion of α-actinin also decreased secretion of vWF from WPBs in endothelial cells [75].

Most likely, contraction is a result of several synergistic mechanisms, including actin polymerization/crosslinking and myosin driven contraction [117,120]. Synergism might be important as the likely isotropic orientation of actin filaments within the spherical geometry of the coat is not optimal for actin-myosin filament sliding [120]. However, based on these observations, it has also been proposed that the main role of actin dynamics in the post-fusion phase might be stabilization rather than contraction [82,88,104,107,125].

The role of coat contraction and vesicle compression seems to be adapted to the physiological setting of secretion. For secretion of bulky, poorly soluble cargo (e.g., pulmonary surfactant, vWF), it is mainly to provide the force necessary to expel cargo from fused granules through a narrow fusion pore [43,47,64,80,81,120] and/or modulate cargo release from individual granules [118]. Whereas in other cells, in particular acinar cells, the actomyosin provides the force necessary to either compress/stabilize fused granules and directly retrieve them from the PM [106,107] or integrate the vesicle membrane against a high hydrostatic pressure into a limited apical PM, so that the membrane can subsequently be retrieved by the cell through compensatory endocytosis [92,121,122].

### 5.2. Endocytosis

Compensatory endocytosis is one of the primary mechanisms through which cells maintain their surface area after exocytosis [126]. It is well established that actin interacts with components of the endocytic machinery and that Arp2/3-mediated actin polymerization contributes to multiple steps during endocytosis [127]. However, one of the first reports directly linking secretory granule exocytosis, actin coating of fused granules and compensatory endocytosis came from *Xenopus laevis* oocytes [106] by an endocytic mechanism known as “kiss-and-coat” [109]. Exocytosed cortical granules were directly retrieved from the PM by F-actin coats that assembled on their surface. The F-actin coat maintained invaginated compartments and drove closure of the exocytic fusion pores. Myo1c coupled polymerizing actin to granule membranes and mediated force production during compensatory endocytosis. So far, there is no evidence for a “kiss-and-coat” mechanism in mammalian eggs; however, compensatory endocytosis in mouse eggs also requires actin cytoskeleton dynamics. A direct involvement of the actin coat is yet to be shown [128].

Similar observations have been made in pancreatic beta and acinar cells. Actin-coating of fused insulin granules leads to local enrichment of endocytic proteins quickly after granule fusion [48] and zymogen granules, at least in part, also seem to be retrieved without collapse into the PM [107,129]. For zymogen granules, however, endocytosis is often delayed, and the granules persist for a long time as “ghosts“, likely to permit compound exocytosis by subsequent fusion of secondary granules to the ghost [129].

In most of the non-neuronal secretory cells where actin coat formation on exocytosed granules has been reported, compensatory endocytosis is not directly driven by actin coat dynamics. In salivary gland acinar cells, the vesicle membrane is integrated into the PM before retrieval by clathrin-independent endocytic mechanisms [130,131]. In endothelial cells, compensatory membrane retrieval was described to depend on clathrin coated pits, which bud from fused WPBs, suggesting immediate retrieval of the membrane after exocytosis [132]. In primary isolated ATII cells, secretory vesicle membrane is integrated into the PM due to contraction of the actin coat. Cell capacitance measurements did not find any immediate compensatory endocytosis [81,133]. Whether this differs from the in vivo situation with restricted apical membrane space is yet unknown. It is interesting to note that direct contribution of actin coating to compensatory endocytosis has so far only been observed in cells where Arp2/3 recruitment to the fused vesicles was observed [48] or predicted [106], but not in cells where F-actin nucleation is Arp2/3 independent and formins have the principal role as nucleators [43,80,111,120]. This suggests that the architecture of the actin coat may define its main purpose-secretion vs. membrane retrieval and/or that actin nucleation is part of the endocytic process as observed in other systems [134,135].

## 6. Conclusions and Outlook

It is well established that actin and myosin are central players for regulated exocytosis in non-neuronal secretory cells. From providing tracks to target secretory granules to fusion sites to actively squeezing bulky cargoes from fused vesicles and retrieving excess membrane following fusion, actin, myosin and a plethora of interacting regulatory lipids and proteins contribute to secretion. Our understanding of these interactions and molecular mechanisms is increasing. However, how this is temporarily and spatially regulated from the pre- to the post-fusion phase is still enigmatic. This is particularly so for how changes in actin polymerization/depolymerization are regulated—from a dense cortical barrier meshwork to a permissive, docking and fusion promoting gel, to a complex actin structure essential for fusion pore stabilization and post-fusion actin coating. Similarly, myosins seem to serve multiple functions, from force generation to filament crosslinking and stabilization of membranes and actin networks. All these processes are intimately intertwined. The advent of novel, super-resolution technologies promises to provide means to follow these interactions with ever increasing detail to increase our understanding of the molecular mechanism governing secretion not only in neuronal but also in non-neuronal secretory cells.

## Figures and Tables

**Figure 1 cells-09-01455-f001:**
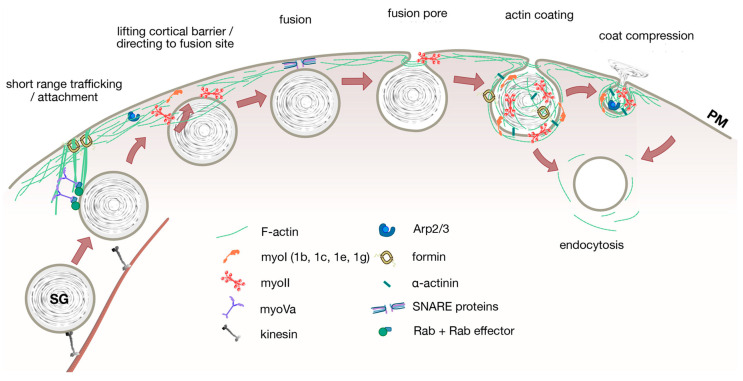
A simplified model of the roles of actin and myosins in non-neuronal exocytosis.The scheme depicts potentially conserved or common mechanisms that have been observed in at least two different secretory cell types. Secretory vesicles are usually transported to the cell periphery on microtubules by motor proteins kinesins. Once in close proximity to the cell cortex, secretory vesicles might be guided towards (and attached to) the cortex by myoV motors along actin bundles attached to the PM. Actin cortex remodeling by myoII and possibly also myoI is necessary to enable access of large secretory vesicles to the PM for fusion of vesicles with the PM. The cortical actin network is essential to direct exocytic vesicles to the fusion site and for stable attachment or docking of granules close to the PM to facilitate fusion via SNARE proteins. F-Actin and myosin also provide force to modulate fusion pore diameters. Actin is polymerized on fused vesicles. Myosins are then recruited to the “actin coat” to either generate compressive forces for active secretion of vesicle contents or integration of vesicle membrane against a high hydrostatic pressure into the PM, likely in conjunction with an actin polymerization/crosslinking mechanism. In some cells, actin coats also stabilize fused secretory vesicles for endocytic retrieval.

**Table 1 cells-09-01455-t001:** Structural and regulatory proteins and lipids found to be involved in pre- and post-fusion stages of exocytosis in various non-neuronal secretory cells. Abbreviations: *D.m.*, *Drosophila melanogaster*; *D.r.*, *Danio rerio*; *H.s.*, *Homo sapiens*; *M.m.*, *Mus musculus*; *O.c.*, *Oryctolagus cuniculus*; *R.n*. *Rattus norvegicus*; *X.l.*, *Xenophus laevis*.

	Pre-Fusion	Post-Fusion
	*Trafficking*	*Cortical attachment/actin remodeling*	*Fusion pore dynamics*	*Actin coat formation, stabilization, contraction*	*Endocytosis*
*α-Actinin*		Muscle cells (*R.n.*) [136]		Endothelial cells (*H.s.*) [75]Alveolar type II cells (*R.n.*) [120]	
*Annexin*		Alveolar type II cells (*R.n.*) [71]Endothelial cells (*H.s.*) [72,73]			
*Arp2/3*		NK cells (*H.s.*) [46]		Salivary gland acinar cells (*D.m.*) [64]Beta cells (*M.m.*) [48]	
*Cdc42*		Beta cells (*M.m.*) [137]		Oocytes (*X.l.*) [106,110]	
*Clathrin*					Beta cells (*M.m.*) [48]Endothelial cells (*H.s.*) [132]
*Cofilin*		Sperm (*H.s.*) [138]		Alveolar type II cells (*R.n.*) [120]	
*DAG*		CT lymphocytes (*M.m.*) [139]		Oocytes (*X.l.*) [110]	
*Dynamin*			Sperm (*M.m.; H.s.*) [140,141]		Beta cells (*M.m.*) [48,142]Endothelial cells (*H.s.*) [132]
*Formins*	Epithelial cells (*D.m.*) [15]Pancreatic acinar cells (*M.m.*) [14]	Mast cells (*M.m.*) [45]	Beta cells (*M.m.*) [48]	Alveolar type II cells (*R.n.*) [80]Beta cells (*M.m.*) [48]Salivary gland acinar cells (*D.m.*) [111]	
*MARCKS*		Airway epithelial cells (*M.m.*) [60]Embryonic cells (*D.r.*) [143]Pancreatic acinar cells (*R.n.*) [144]			
*Melanophilin*	Melanocytes (*M.m.*) [29,145,146]	Melanocytes (*M.m.*) [29]			
*MLCK*		Renin-secreting cells (*M.m.*) [51]		Alveolar type II cells (*R.n.*) [120] Salivary gland acinar cells (*M.m.*) [119]	
*MyoI*		Fibroblasts (*M.m.*) [52]Alveolar type II cells (*R.n.*) [53]B lymphocytes (*M.m.*) [54]		Alveolar type II cells (*R.n.*) [53]Oocytes (*X.l.*) [123]Oocytes (*X.l.*) [117]	Oocytes (*X.l.*) [123]
*MyoII*		NK cells (*H.s.*) [46]Renin-secreting cells (*M.m.*) [51]Endothelial cells (*H.s.*) [147]	Pancreatic acinar cells (*M.m.*) [88]	Alveolar type II cells (*R.n.*) [120]Salivary gland acinar cells (*M.m.; D.m.*) [108,111]Oocytes (*X.l.*) [117]Lacrimal gland acinar cells (*O.c.*) [47]Endothelial cells (*H.s.*) [43,147]	
*MyoV*	Endothelial cells (*H.s.*) [26]Melanocytes (*M.m.*) [29,145]Epithelial cells (*H.s.*) [148]Beta cells (*M.m.*) [66]	Endothelial cells (*H.s.*) [3] Beta cells (*M.m.*) [66]Mast cells (*M.m.*) [44]			
*MyoVII*		Beta cells (*M.m.*) [68]			
*MyRIP (Exophilin 8)*	Endothelial cells (*H.s.*) [12,26]	Endothelial cells (*H.s.*) [12,26]Beta cells (*M.m.*) [68]			
*PIP* _2_		CT lymphocytes (*M.m*) [139,149]		Salivary gland acinar cells (*D.m.*) [64]	
*PKC*				Oocytes (*X.l.*) [110]Endothelial cells (*H.s.*) [118]	
*Rab3*	Sperm (*H.s.*) [28]Endothelial cells (*H.s.*) [72]Oocytes (*M.m.*) [150]	Epithelial cells (*H.s.*) [69]Sperm (*H.s.*) [151]	Sperm (*H.s.*) [94]		
*Rab11*	Epithelial cells (*H.s.*) [148]Oocytes (*M.m.*) [27]	Mast cells (*M.m.*) [77]			
*Rab27*	Endothelial cells (*H.s.*) [26]Oocytes (*M.m.*) [27]Sperm (*H.s.*) [28]Melanocytes (*M.m.*) [29,145]	Endothelial cells (*H.s.*) [26]Sperm (*H.s.*) [28,151]Oocytes (*M.m.*) [150]Melanocytes (*M.m.*) [29]CT lymphocytes (*M.m.*) [30,149]Mast cells (*M.m.*) [44] Neutrophils (*M.m.*) [152]Beta cells (*M.m.*) [153]			
*Rho*		Mast cells (*R.n.*) [154]		Oocytes (*X.l.*) [117]Alveolar type II cells (*R.n.*) [80] Salivary gland acinar cells (*D.m.*) [111]Pancreatic acinar cells (*M.m.*) [107]Endothelial cells (*H.s.*) [155]	
*ROCK*				Alveolar type II cells (*R.n.*) [120]	
*SLP4 (Granuphilin)*		Endothelial cells (*H.s.*) [156]Epithelial cells (*H.s.*) [69,148]			
*Tropomyosin*				Salivary gland acinar cells (*M.m.*) [157]	
*UNC-45*		NK cells (*H.s.*) [50]			
*WASP*				Salivary gland (*D.m.*) [64]Oocytes (*X.l.*) [106]	
*Zyxin*		Endothelial cells (*H.s.*) [75]		Endothelial cells (*H.s.*) [75]

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
