# Peer review of "Actin and Myosin in Non-Neuronal Exocytosis"

_cells, 2020, doi:10.3390/cells9061455_

Round 1
Reviewer 1 Report
This review by Pika Miklavc and Manfred Frick is focussed on the role of actin and myosin in non-neuronal exocytosis. They highlight the role of cytoskeletal elements during the different phases of vesicle exocytosis and propose a general molecular mechanism of exocytosis in non-neuronal cells. Moreover they list central cytoskeletal elements and regulatory molecules identified in the different non-neuronal cells.
Generally, this review well-written and comprehensive could represent an important source of information for researchers working in the field of exocytosis in non-neuronal cells. However, I have comments on the Table1. I don't understand how the authors selected proteins and lipids presented in the table. Why actin and SNARE proteins do not appear in the Table 1. For clarity the authors could group the different proteins by family. Finally, a part of a line is missing. Concerning the References section, the volume and pages are missing for a large number of references thus the authors must complete these references.
I have also some minor comments:
line 60 centre should be replace by center
line 94 a space must be added between Von and Willebrand
line 196 annexin 7 should be replace by annexin A7
line 129 B should be added to Latrunculin
line 131 Abbreviation of natural killer NK should be added here and not line 137
line 288 to line 291 Myosin II should be replace by MyoII
line 312 favorable should be replace by favourable
line 359 and line 798 “and secretion” is not necessary and could be remove
line 376 “s” should be added to Reference
line 541 Suedhof should be replace by Südhof
line 771 reference 153 is not aligned
Reviewer 2 Report
This is an informative and timely review summarizing available information in a quickly developing field, roles of the actin cytoskeleton in exocytosis. Considering history of various conceptual shifts that occurred in this field this review will be of great help to readers. However, I recommend several corrections and clarifications, as specified below.
Neither figure nor table is cited in the main text.
p. 1, l. 17: The term “lift the barrier”, which is used in the abstract, figure legend and throughout the manuscript. is very confusing. I assume that the real meaning is that the cortex locally disassembles to allow the secretory vesicle to access the plasma membrane. “Lifting” does not imply any disassembly to my understanding.
p. 2, l. 50: Why is it difficult to secrete hydrophilic cargos, as the context of the sentence suggests? I would think that it is more troublesome to secrete hydrophobic cargo.
p. 3. l. 97-98: Given that MyRIP is not a motor, its binding to actin cannot regulate “trafficking” better than myoVa motor, as stated here. The cited paper shows that MyRIP “restricts” cargo movement.
p. 3, l. 100: Specify which myosin class is referred to in this sentence.
p. 3. l. 129: Specify which formin is referred to in this sentence.
p. 3, l. 131-132: In the cited study, the FRAP turnover halftimes were decreased, which indicated higher turnover rates. The statement in the manuscript says opposite - increased times, which would indicate slower rates.
p. 3, l. 138-140: The cited paper #46 showed that actin “dynamism” (the term used in that paper) depended on both Arp2/3 and myoII. It is not accurate to say that they “showed displacement of branched cortical actin filaments during exocytosis, which was dependent of myoII [46].” Given that branched networks are less permissive for myosin binding, myosin contribution most likely involved unbranched actin filaments. I suggest to delete "branched" and acknowledge the dual mechanism.
p. 3, l. 144-145: This sentence rather belongs to the previous paragraph, because there is nothing about myosin II in the cited experiment.
p. 4, l. 147-149: It is hard to imagine how secretion can be “mediated by myoII phosphorylation”. May be, “negatively regulated” or similar?
p. 4, l. 158-159: References 56 and 57 citing PIP2 reviews are pretty old. Are there newer ones?
p. 4, l. 165: A word is missing in the sentence.
p. 5, l. 221: Specify which myosin class is referred to in this sentence. All cited references are about myosin II.
p. 5, l. 222: What is meant by “among others” – other mediators, other vesicles or other cells?
p. 7, l. 306-307: Move the sentence citing ref. 75 after the last sentence of the paragraph, which cites ref. 121, otherwise it is not clear which paper showed what the beginning of the paragraph describes.
p. 7, l. 309-314: In this paragraph, the authors appear to completely dismiss the role of myosin-driven contraction, even though other parts of the manuscript, including the following paragraph, properly acknowledge it. This paragraph needs to be in line with the fact that contraction does play a role.
p. 7, l. 326: To my knowledge, Arp2/3-dependent actin polymerization, rather than “actomyosin”, plays a role in endocytosis, as also reviewed in the cited reference 128.
p. 19, l. 803-804: The figure legend says that myoI is involved in cortex remodeling along with myoII. I agree about myoII, but no evidence is provided for myoI, either in the main text or in the legend. I doubt that such evidence exists, but would like to know if it does.
p. 20: Table lists the proteins in no obvious order. I suggest organizing them by functions, so that motors were together, as well as GTPases, and so on.
